# Long COVID Complicated by Fatal Cytomegalovirus and Aspergillus Infection of the Lungs: An Autopsy Case Report

**DOI:** 10.3390/v15091810

**Published:** 2023-08-25

**Authors:** Lucia Krivosikova, Tereza Kuracinova, Peter Martanovic, Michaela Hyblova, Jozef Kaluzay, Alexandra Uhrinova, Pavol Janega, Pavel Babal

**Affiliations:** 1Institute of Pathological Anatomy, Faculty of Medicine, Comenius University in Bratislava, 814 99 Bratislava, Slovakia; lucia.krivosikova@fmed.uniba.sk (L.K.); terezka.kuracinova@gmail.com (T.K.); pavol.janega@fmed.uniba.sk (P.J.); 2Health Care Surveillance Authority, 829 24 Bratislava, Slovakia; peter.martanovic@udzs-sk.sk; 3Medirex Group Academy, n.p.o., 949 05 Nitra, Slovakia; michaela.hyblova@medirex.sk; 44th Department of Internal Medicine, Comenius University in Bratislava, 814 99 Bratislava, Slovakia; kaluzay@gmail.com; 5Agel Radiologia, s.r.o., 811 06 Bratislava, Slovakia; uhrin.alexa@gmail.com; 6Centre of Experimental Medicine, Institute of Normal and Pathological Physiology, Slovak Academy of Sciences, 813 71 Bratislava, Slovakia

**Keywords:** COVID-19, long COVID, immune dysregulation, corticoid therapy, cytomegalovirus, *Aspergillus*

## Abstract

After the acute phase of COVID-19, some patients develop long COVID. This term is used for a variety of conditions with a complex, yet not fully elucidated etiology, likely including the prolonged persistence of the virus in the organism and progression to lung fibrosis. We present a unique autopsy case of a patient with severe COVID-19 with prolonged viral persistence who developed interstitial lung fibrosis complicated by a fatal combination of cytomegalovirus and *Aspergillus* infection. SARS-CoV-2 virus was detected at autopsy in the lungs more than two months after the acute infection, although tests from the nasopharynx were negative. Immune dysregulation after COVID-19 and the administration of corticoid therapy created favorable conditions for the cytomegalovirus and *Aspergillus* infection that were uncovered at autopsy. These pathogens may represent a risk for opportunistic infections, complicating not only the acute coronavirus infection but also long COVID, as was documented in the presented case.

## 1. Introduction

COVID-19 is a viral disease caused by SARS-CoV-2, affecting mainly the respiratory system. The typical microscopic picture of COVID-19 pneumonia includes diffuse alveolar damage (DAD), which starts with pulmonary capillaritis, predominantly lymphocytic interstitial infiltrate, the damage of alveolar epithelial cells, hyaline membrane formation, thrombosis and hemorrhages [1]. Squamous metaplasia and various levels of organization can be observed in the later stages. The immune system shows signs of dysregulation, combining features of hyperinflammation and immunosuppression. Hyperinflammation is exhibited by highly activated innate immune cells and the elevated expression of inflammatory mediators [2]. COVID-19-associated immunosuppression manifests by lymphopenia, which is predictive of poor outcomes and infectious complications [3]. 

Cytomegalovirus (CMV) infection is asymptomatic in most people. However, it can cause severe complications in immunocompromised patients. Latent CMV infection was recognized as a factor associated with an increased risk of COVID-19-related hospitalization [4]. Low lymphocyte count in patients with COVID-19 is indicative of possible CMV infection development [5].

*Aspergillus* infection is another frequent complication in immunocompromised patients, including those with COVID-19 [6]. This fungus affects a high percentage of patients with acute COVID-19 hospitalized in the intensive care units [7,8].

Long COVID is an imprecise term used for various conditions and symptoms. The etiology of the symptoms is therefore complex and variable, but persisting infection, the production of autoantibodies, microthrombi and lung fibrosis are among the major factors [9]. The cases of COVID-19 with coinfection by multiple agents described in the literature are mostly clinical reports and most cases reporting CMV infection are based on serology results. Here, we present an autopsy case of persisting COVID-19 pneumonia complicated by simultaneous large-scale CMV and *Aspergillus* affection of the lungs. Beside extensive fibrosis and signs of acute DAD, we were able to detect SARS-CoV-2 virus in the lungs even more than two months after the acute infection. CMV infection showed the typical morphology and was confirmed by immunohistochemistry. The presented case documents the increased risk of serious complications of SARS-CoV-2 infection in patients with preexisting pulmonary conditions like bronchial asthma, as was the true in the case of our patient. 

## 2. Case Presentation

### 2.1. Clinical Course

A 64-year-old man with a history of bronchial asthma, on chronic inhalation therapy with salmeterol and fluticasone, was infected by the SARS-CoV-2 virus. He developed severe COVID-19, requiring prolonged hospitalization. After one month, he was successfully discharged from the hospital. During his recovery at home, he was brought to the emergency department because of a sudden worsening of dyspnea. The dyspnea progressed rapidly and led to the death of the patient. The clinical history is summarized in the following text.

The patient started to show symptoms on 1 February 2021. COVID-19 manifested in a fever up to 38.5 °C and SARS-CoV-2 infection was confirmed by a rapid antigen test and by RT-PCR test. Initially, he was treated on an outpatient basis by azithromycin for six days, ipratropium bromide, nadroparin in prophylactic dose and prednisone 20 mg/day. Two weeks later, the patient was brought to hospital because of weakness, progression of dyspnea and sO_2_ of 80%. A CT pulmonary angiogram confirmed low-risk peripheral pulmonary embolism. The extent of ground-glass opacity (GGO) was estimated at 75% of the total lung volume (Figure 1). There were no significant abnormalities in the standard laboratory tests (CRP 88 mg/L, normal NT-proBNP). The initial treatment in the hospital included oxygen therapy via face mask, nadroparin in full therapeutic dose, dexamethasone 8 mg/day, ceftriaxone and symptomatic and supportive treatment. Moxifloxacin was added a few days after admission because of an increase in CRP (129 mg/L). The sO_2_ was not improving and the patient required high-flow nasal oxygen therapy (HFNO), and the dose of dexamethasone was increased to 12 mg/day. According to local protocol, remdesivir was not administered because of IgM positivity. Monoclonal antibodies were not available.

After eleven days of hospitalization, CRP stabilized at a normal level and antibiotics were stopped. HFNO withdrawal was possible after two weeks. Dexamethasone was reduced to 4 mg/day, but bilateral infiltration was still present on the chest X-ray. The hospitalization was complicated by a moderate elevation of CRP (46 mg/L), requiring a second course of antimicrobial treatment with meropenem and fluconazole. No specific agent was cultivated from sputum. 

The patient was discharged after one month without clinical signs of infection. A chest CT scan demonstrated diffuse reticular changes suggesting the development of lung fibrosis (Figure 2). The pulmonary consultant recommended long-term treatment with dexamethasone 4 mg/day. The patient was fully mobile at the time of discharge, with a normal sO_2_ at rest. A marked desaturation often occurred after light physical activity with a rapid improvement at rest.

A few weeks later, the patient required a three-day hospitalization because of thrombosis of the femoral artery. An angiology consultant changed the anticoagulation therapy from nadroparin to apixaban.

Two weeks after this episode, 68 days after the initial diagnosis, the patient was brought to the hospital again due to worsened dyspnea. Rapid PCR tests conducted from the nasopharyngeal swab for COVID-19, Influenza A, and Influenza B were negative. During the transfer of the patient to the emergency department, respiratory failure and shock rapidly progressed within a few minutes. Intensive resuscitation was not successful. Most laboratory tests were in norm, D-dimer 1.58 mg/L; CRP 85.9 mg/L; sO_2_ 59.3.

### 2.2. Autopsy Investigation

An autopsy was performed 4 h after death. From the outer inspection we observed that the skin was of waxy white color, without efflorescence and peeling, the nutritional status was appropriate and the extremities were without edema. From the inner inspection we report that the head was not opened due to the request of relatives. The pleural cavity was without excessive fluid, and the pleura was smooth and transparent. The left lung weighed 490 g, and the right lung weighed 590 g. The lungs were collapsed, of dark reddish-pink color, of elastic to hard elastic consistency and with a rough to bumpy surface. The cut surface showed marble drawings with alternating grayish-white and dark-red areas. Upon applying pressure, turbid reddish to grayish, slightly foamy fluid came out of the tissue. The pulmonary arteries contained liquid blood. The heart was 12 cm × 11 cm × 6 cm large, the circumference of the valves was normal, the right ventricle was slightly dilated with flattened trabeculae, the endocardium was smooth and the myocardium wall thickness was 3 mm right, 15 mm left. The intimal surface was covered with white to yellow plaques by 40% and 50% in the coronary arteries and the aorta, respectively. The mucosa of the stomach and intestines was smooth, grayish-white and without defects. The pancreas was of a granular structure, a light-beige color and with 2–4 mm large white spots in the adjacent adipose tissue. The liver was of an appropriate size, with a smooth surface, red-brown and fine-granular on the cut surface. The spleen was of an appropriate size (110 g), so were the kidneys, with a normal structure on the section surface.

#### Histological Findings

A microscopic examination of the lung tissue showed two different alternating patterns. The first pattern of the acute phase of diffuse alveolar damage consisted of alveolar capillaritis with lymphocytes and neutrophils in the interstitium, denuded alveolar epithelium, alveolar edema and hyaline membrane formation (Figure 3). The other pattern of the organizing phase, covering more than 50% of the tissue, showed signs of organization, fibrosis and thickened alveolar septae, with variable intensity of lymphocytic infiltrate. Focally, alveolar epithelium had signs of hyperplasia and hypertrophy appearing as cuboidal epithelium (Figure 3). 

Focally, there were areas with necrosis, neutrophilic infiltration and colonies of Y-shaped hyphae, as is typical for *Aspergillus.* The fugus was spreading from bronchial lumens into parabronchial parenchyma (Figure 4); vascular infiltration was not observed. In the alveolar epithelium, there were diffusely dispersed conspicuous large cells with nuclei with prominent intranuclear dense inclusion. Intra-nuclear inclusion bodies of these enlarged cells lead to an owl’s eye appearance, thus allowing the direct detection of CMV infection by histopathology evaluation. Accordingly, these nuclei stained positive with anti-cytomegalovirus antibodies. Staining with anti-SARS-CoV-2 nucleocapsid protein antibody detected diffusely dispersed individual or small clusters of alveolar epithelial cells with cytoplasmic positivity (Figure 4). The distribution of cells infected by both types of viruses was diffuse, including areas with signs of diffuse alveolar damage and areas with progressing fibrosis.

The mucosa of the gastrointestinal tract was without remarkable inflammatory infiltration. There was venostasis in the liver and mild arteriosclerotic nephrosclerosis in the kidneys. The white pulp of the spleen was without signs of activation. There was venostasis in the red pulp with increased numbers of plasma cells and neutrophils, indicating the infectious activation of the spleen. A histology of the pancreas showed lipomatosis, focal interstitial fibrosis and focal necrosis of the parenchyma and of the adipose tissue, corresponding to the acute exacerbation of chronic pancreatitis. There was moderate myoadenomatous hyperplasia of the prostate. An immunohistochemical investigation of the tissues of the heart, liver, intestines, pancreas and the kidneys did not detect positivity with the anti-SARS-CoV-2, nor with the anti-CMV antibodies.

The immunohistochemical staining of formalin-fixed, paraffin-embedded material was performed with anti-SARS-CoV-2 nucleocapsid protein prediluted antibody (BioSB, Santa Barbara, CA, USA) and anti-CMV Flex prediluted antibody, and the reaction was visualized using the Flex EnVision immunohistochemical detection system (Agilent Technologies, Santa Clara, CA, USA) in the DAKO Autostainer (DAKO, Glostrup, Denmark).

## 3. Discussion

COVID-19 represents, in the majority of cases, a virosis with a mild clinical course. However, 4–5% of patients infected with SARS-CoV-2 develop a more severe disease, which requires hospitalization. This includes pneumonia with hypoxia, often with systemic thrombosis, cardiac injury or renal failure [10]. The severity and the course of the infection depends on multiple factors, including age, sex, ethnicity, comorbidities and lifestyle [11], with genetic predispositions probably playing a crucial role [12]. Two months after the initial diagnosis of SARS-CoV-2 infection, the rapid PCR test from the nasopharyngeal swab was negative upon admission to the emergency department. False negativity is a well-described phenomenon that can be caused by multiple factors in all levels of the process [13]. Moreover, in patients with prolonged clinical course, the virus might survive in the sputum longer than in samples from the upper respiratory tract [14]. However, because of the abrupt clinical course during the last hospitalization, there was not enough time to repeat the test or to take a different sample. 

Nevertheless, the postmortem immunohistochemical analysis identified an extensive positivity of the virus in the lung tissue, which indicated persistent SARS-CoV-2 pulmonary infection. This was much longer than the 3 or 4 weeks reported in other studies [15,16]. In some patients with long COVID, SARS-CoV-2 infection in the gut persisting for several months serves as a reservoir of the virus [17]. In the presented case, immunohistochemistry did not uncover the SARS-CoV-2 virus in any other tissue except the lungs, which might represent a reservoir of the infection as well. Our patient suffered from bronchial asthma, a comorbidity with no increased risk for unfavorable outcomes of COVID-19 compared to the general population [18]. Long COVID, in turn, is considered an adverse prognostic factor in patients with bronchial asthma [19]. In our patient, COVID-19 induced massive fibrotic change in the pulmonary parenchyma. 

Soon after the COVID-19 pandemic took over, it became clear that this disease seriously disturbs the normal functions of the immune system [20]. Immunosuppressive drugs like corticoids are proven to reduce hyperinflammation and mortality rates and increase ventilator-free days in patients with severe COVID-19 [21]. However, combined with COVID-19-related lymphopenia and antibiotic-treatment-induced microbial imbalance, it provides an ideal condition for opportunistic infections [22]. COVID-19 patients were described with all kinds of opportunistic infections, including fungi, viruses, bacteria, protozoa and helminths [21].

Our patient developed an *Aspergillus* infection, which is the leading cause of opportunistic infections in COVID-19 patients. In a like manner, aspergillosis was described as the leading complication of patients with severe influenza [23]. Patients with COVID-19-associated pulmonary aspergillosis (CAPA) require admission to an intensive care unit in more than 95% of cases, with a mortality rate over 50% [7]. Most of the described cases of CAPA are based on clinical diagnosis, which might not represent an invasive disease. Invasive CAPA is a rare finding in autopsy series, found in less than 2% of cases [24]. However, autopsy studies show that the burden of coinfections, including CAPA, is often underestimated clinically, compared to autopsy findings [25]. In our case, the diagnosis of CAPA was based only on the autopsy finding. Clinically, the microbiological tests did not show any signs of *Aspergillus* colonization or infection. Indirect tests, such as the galactomannan test, were not standards of care in our region. 

Relevant underlying factors, such as diabetes mellitus, smoking, hypertension disease and pulmonary diseases, do not represent principal risk factors for CAPA, although they represent factors with negative prognostic outcomes [8]. Several studies point to the immunomodulating therapy applied in COVID-19 patients as playing a critical role in the development of CAPA. It has to be considered that, since the application of corticosteroids and interleukin-6 blockers (mainly tocilizumab) became a standard treatment for COVID-19, such medication could be associated with the high incidence of CAPA [26]. Typical radiological findings in invasive Aspergillosis are the presence of cavitations and bronchiectasis [27], the second of which were observed in the CT scans in our case. It is also likely that our patient developed aspergillosis later in the clinical course, not during hospitalization in ICU. In such a case, the most important underlying conditions would be the long-term corticoid therapy and prolonged COVID-19.

In our report, this severe condition was complicated by another infectious agent. CMV usually causes latent infection, with 80% seropositivity in older patients. Combined immune dysregulation in COVID-19 patients might lead to the reactivation of a latent CMV infection to a symptomatic one [21]. It is not surprising that CMV seropositivity is associated with an increased chance of developing a severe course of COVID-19 [28]. In our patient, the CMV infection was only detected post-mortem at autopsy. The diagnosis can be made through the histopathological identification of characteristic large nuclei and confirmed by immunohistochemical staining, mainly in the alveolar epithelial cells [29]. 

Reports of concomitant infections by multiple pathogens, including opportunistic infections in COVID-19 patients, are predominantly based on clinical diagnosis [30]. An autopsy case similar to ours was reported, describing a 68-year-old diabetic man who was admitted to hospital for COVID-19 pneumonia, discharged and then re-admitted with Klebsiella pneumoniae infection with empyema. A few days before death, he acquired infections of *Aspergillus*, CMV and mucormycosis, which were confirmed by autopsy findings. However, in this case, the authors do not mention SARS-CoV-2 positivity or signs of COVID-19 pneumonia at the time of autopsy [31].

In our case, the persisting COVID-19 disease in a patient with bronchial asthma led to the development of post-COVID lung fibrosis. This condition, in combination with corticoid therapy, was complicated by two additional serious infections, CMV and *Aspergillus*, which resulted in terminal respiratory failure.

## Figures and Tables

**Figure 1 viruses-15-01810-f001:**
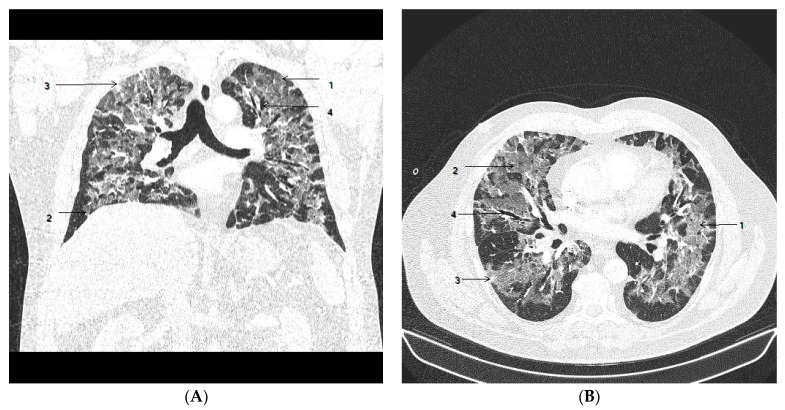
Chest CT on admission to hospital. Coronal view (**A**) and axial view (**B**) show areas of ground-glass opacities (GGO) (1) with more than 75% involvement of the lung volume, crazy paving (2) and subpleural consolidation (3) were observed, with mild bronchiectasis (4) in segmental and subsegmental distribution.

**Figure 2 viruses-15-01810-f002:**
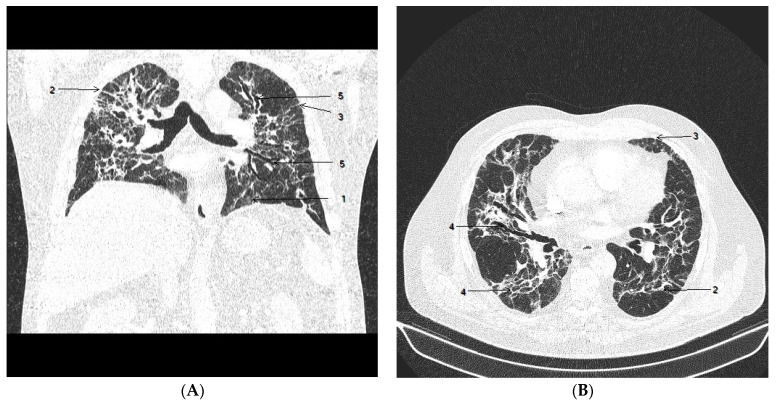
Chest CT after 4 weeks of hospitalization. Coronal view (**A**) and axial view (**B**) show gradual resolution of GGO (1); there were multiple subpleural bands and multifocal linear abnormalities (2), interlobular septal thickening (3) with minimal lung architectural distortion (4) and minimal progression of bronchiectasis (5), predominantly in the right middle and the left upper lobe.

**Figure 3 viruses-15-01810-f003:**
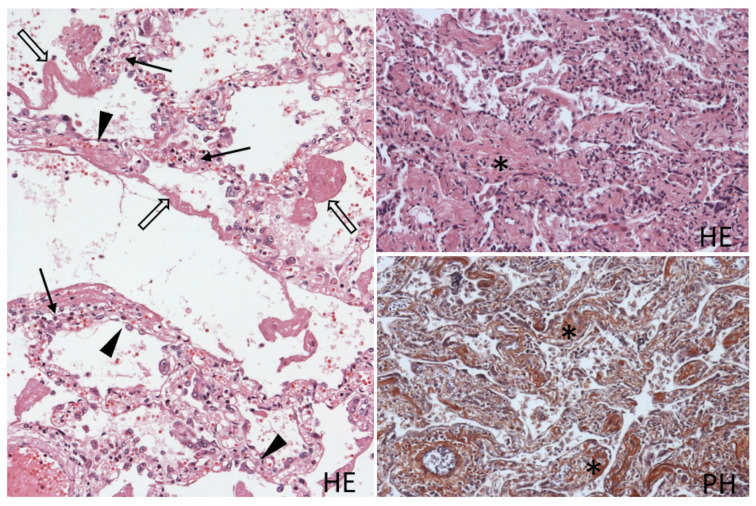
Histological picture of the lung tissue with alternating areas of the acute phase with diffuse alveolar damage (**left**) and areas of organization with interstitial fibrosis (**right**). Alveolar capillaritis (arrow), detached hypertrophic alveolar epithelium (arrowhead), hyaline membranes (open arrow) lining the alveoli, thickened alveolar septae with fibrosis (asterisk), with brown staining of collagen in PH staining. Hematoxylin and eosin (HE), 100×, phosphotungstic acid hematoxylin (PH); 100×.

**Figure 4 viruses-15-01810-f004:**
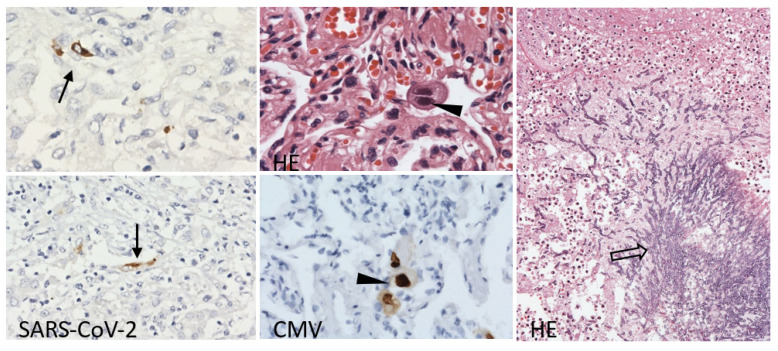
Lung tissue with persistent scattered SARS-CoV-2 nucleocapsid protein positivity in the alveolar lining (arrow), large alveolar epithelia with large intranuclear inclusions with cytomegalovirus protein-positive nuclei (arrowhead) and *Aspergillus* hyphae spreading into parabronchial pulmonary parenchyma (open arrow). Immunoperoxidase technique, diaminobenzidine 200×, 100×; Hematoxylin and eosin (HE), 400×, 100×.

## Data Availability

No new data were created or analyzed in this study. Data sharing is not applicable to this article.

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
