# Peer review of "Long COVID Complicated by Fatal Cytomegalovirus and Aspergillus Infection of the Lungs: An Autopsy Case Report"

_viruses, 2023, doi:10.3390/v15091810_

Round 1
Reviewer 1 Report
This is an interesting case report of a patient succumbing to a combined superinfection with cytomegalovirus and aspergillus of the lung in persistent COVID-19. Although clearly of interest as a well documented case of unusual triple infection, the paper shows some weaknesses which need to be addressed.
Given the fact that complete autopsy was performed, the description of the findings is rather limited and needs to be more detailed. How was the precise distribution of the different infectious agents in the lung, especially in relation to the two patterns described, namely acute DAD versus progressing fibrosis? How was the distribution of the two viruses? Intermingled or separate? Was the area with florid DAD associated with any of the viruses or aspergillosis? Did aspergillus result in typical angioinvasion with infarction?
No mention is made about extrapulmonary findings. This needs to be added. Specifically the authors need to comment whether any of the three infectious agents were found in other organs, also using immunohistochemistry.
Minor points:
Given that this is an autopsy report, the relevant autopsy papers, also concerning superinfections of COVID-19 should be cited and discussed briefly.
“Long COVID” is a rather imprecise term used for a variety of conditions. This case is best classified as severe COVID-19 with prolonged viral persistence.
Figure 1B is of poor quality and should be replaced to better show the progressive fibrosis, perhaps also using Masson trichrome stain or similar. In addition, owls eye cells should also be shown in HE.
The discrepancy between negative throat swap and detection of the virus in lung tissues is well described especially for severe COVID-19 with prolonged course. Please mention and use appropriate reference.
The English language is poor in several passages and needs to be improved.
Author Response
We would like to thank the rewiever for the comments, we find them very helpful for improvement of the quality of the paper. P. Babal
Reviewer 1
-Given the fact that complete autopsy was performed, the description of the findings is rather limited and needs to be more detailed. How was the precise distribution of the different infectious agents in the lung, especially in relation to the two patterns described, namely acute DAD versus progressing fibrosis? How was the distribution of the two viruses? Intermingled or separate? Was the area with florid DAD associated with any of the viruses or aspergillosis? Did aspergillus result in typical angioinvasion with infarction?
We added all the requested information in lines 141-192.
-No mention is made about extrapulmonary findings. This needs to be added. Specifically the authors need to comment whether any of the three infectious agents were found in other organs, also using immunohistochemistry.
Added in lines 189-192
Minor points:
-Given that this is an autopsy report, the relevant autopsy papers, also concerning superinfections of COVID-19 should be cited and discussed briefly.
-“Long COVID” is a rather imprecise term used for a variety of conditions. This case is best classified as severe COVID-19 with prolonged viral persistence.
We agree. But since it´s a popular term, we decided to keep it and mention your point in the abstract and introduction.
-Figure 1B is of poor quality and should be replaced to better show the progressive fibrosis, perhaps also using Masson trichrome stain or similar. In addition, owls eye cells should also be shown in HE.
We changed the HE picture with fibrosis and added phosphotungstic acid hematoxylin stain for collagen emphasis (Figure 3); we added CMV infected cells in HE (Figure 4)
-The discrepancy between negative throat swap and detection of the virus in lung tissues is well described especially for severe COVID-19 with prolonged course. Please mention and use appropriate reference.
We included this information in discussion in lines 240-245
Reviewer 2 Report
The case reporte is good, it did not have methodological o clinical failures.
Just I had some questions to the authors.
1) The patient show skin lesions, like Osler nodules or Jenaway lesions or splinter hemorrhage in fingers?
2) The patient show any Roth's lesion?
3) The patient development any endocarditis sing?
4) The patient had an increase in the biochemical parameters related with thrombosis?
5) The patient had data consistent with pulmonary microembolisms?
non
Author Response
We would like to thank the rewiever for the comments, we find them very helpful for improvement of the quality of the paper.
P. Babal
Reviewer 2
1) The patient show skin lesions, like Osler nodules or Jenaway lesions or splinter hemorrhage in fingers?
2) The patient show any Roth's lesion?
3) The patient development any endocarditis sing?
The patient did not show any clinical signs or autopsy findings consistent with endocarditis. We added descriptions of autopsy findings on the heart, endocardium, skin, etc. to make it clear there were no such findings.
4) The patient had an increase in the biochemical parameters related with thrombosis?
5) The patient had data consistent with pulmonary microembolisms?
The patient had terminally mild elevation of D-dimer (line 105), at autopsy no signs of embolism were found.
Reviewer 3 Report
Dear Authors
I would like to thank you for the opportunity of reviewing this interesting paper that is focused on a very remarkable and challenging topic that is a lively argument also in the daily clinical practice. Although it has been more than 2 years since the first outbreak, the coronavirus disease 2019 (COVID-19) pandemic is still having a profound and devastating impact on global healthcare systems. The present manuscript reports an interesting case of histologically-proven long COVID-19 pneumonia complicated with simultaneous large-scale infection by CMV and Aspergillus.
This paper is pleasurable to read, although it suffers from some limitations that Authors can easily adjust in order to slightly improve their review making it more eligible for this important Journal. Furthermore, the Authors can improve some sections of the paper, adding information and including other important references about this topic that, in my opinion, should be cited and discussed.
Although the title is clear and direct, I suggest adding that it is an autopsy case.
In my opinion, the first part of the introduction is a bit unclear. I suggest directly presenting the case in chronological order, possibly avoiding being so schematic (i.e., day 1.., day 2…). Sometimes, well-written prose can help engage the attention of the readers.
In addition, since bacterial and fungal co-infections and superinfections are frequent in COVID-19 patients, especially those severely ill, can the Authors please state if during both hospitalizations of the patient any microbiological tests were performed and if they resulted negative for the presence of other fungal/bacterial/viral microorganisms? In particular, could the Authors please state if blood samples underwent galactomannan test for the detection of aspergillosis infections? [Lancet Respir Med. 2018 Oct;6(10):782-792. doi: 10.1016/S2213-2600(18)30274-1] [Am J Respir Crit Care Med. 2012 Jul 1;186(1):56-64. doi: 10.1164/rccm.201111-1978OC]
Furthermore, please provide the images of the CT examinations described in the text (both the one where COVID-19 pulmonary pattern is evident and the other one with the pulmonary embolism).
In my opinion, the role of CT in COVID-19 patients, especially those who were severely ill, is very important and should be underlined. In fact, it is true that chest CT is one of the main techniques to assess the severity of the pneumonic infection and help detect any complications, suggesting further laboratory investigations. In particular, the presence of consolidations, cavitations, and bronchiectasis should be warning signs for radiologists, since they are associated with the presence of bacterial and/or fungal co-pathogens. In particular, a recent study demonstrated that the presence or the de-novo appearance or the volumetric increase in bronchiectasis is associated with Aspergillus colonization in COVID-19 patients [doi: 10.3390/diagnostics12071617]. Please cite this article and introduce these important aspects in the discussion section, describing if this imaging feature was observed also in the patient. However, as a recent study demonstrated, premortem studies underestimated the real burden of fungal and bacterial co-infections in COVID-19 patients compared to postmortem studies [doi: 10.3390/pathogens12070932].
Chapters 2 and 3 should be merged and “Case presentation” should be used as the title.
Lines 140-141 seem to miss some references (possible suggestions include, doi: 10.3390/diagnostics12040846 and doi: 10.3390/diagnostics11112071).
Line 165, as suggested above, explain if CAPA clinical diagnosis was excluded in your case and how it was performed.
Please better explain why the Authors believe this is an interesting and unique case deserving of publication (for example, it should be mentioned in the conclusion that the persistence of SARS-CoV-2 in the tissues was demonstrated) and make it more engaging for the readers. Also, add a brief sentence explaining the importance of this report at the end of the introduction.
Finally, references do not reflect the style shown in “Authors Guidelines”. (Author 1, A.B.; Author 2, C.D. Title of the article. Abbreviated Journal Name, Year, Volume, page range). Please, format them accordingly.
Best regards,
Author Response
We would like to thank the rewiever for the comments, we find them very helpful for improvement of the quality of the paper.
P. Babal
Reviewer 3
-Although the title is clear and direct, I suggest adding that it is an autopsy case.
We added this information.
-In my opinion, the first part of the introduction is a bit unclear.
We rewrote the first part of the introduction.
-I suggest directly presenting the case in chronological order, possibly avoiding being so schematic (i.e., day 1.., day 2…). Sometimes, well-written prose can help engage the attention of the readers.
We accept this comment and rewrote this part.
-In addition, since bacterial and fungal co-infections and superinfections are frequent in COVID-19 patients, especially those severely ill, can the Authors please state if during both hospitalizations of the patient any microbiological tests were performed and if they resulted negative for the presence of other fungal/bacterial/viral microorganisms? In particular, could the Authors please state if blood samples underwent galactomannan test for the detection of aspergillosis infections? [Lancet Respir Med. 2018 Oct;6(10):782-792. doi: 10.1016/S2213-2600(18)30274-1] [Am J Respir Crit Care Med. 2012 Jul 1;186(1):56-64. doi: 10.1164/rccm.201111-1978OC]
Besides the tests already mentioned in the text, there were no additional microbiological tests performed. Galactomannan test wasn´t in the standards of care in our region - we added this information in the discussion and are citing some of the above mentioned papers.
-Furthermore, please provide the images of the CT examinations described in the text (both the one where COVID-19 pulmonary pattern is evident and the other one with the pulmonary embolism).
In my opinion, the role of CT in COVID-19 patients, especially those who were severely ill, is very important and should be underlined. In fact, it is true that chest CT is one of the main techniques to assess the severity of the pneumonic infection and help detect any complications, suggesting further laboratory investigations. In particular, the presence of consolidations, cavitations, and bronchiectasis should be warning signs for radiologists, since they are associated with the presence of bacterial and/or fungal co-pathogens. In particular, a recent study demonstrated that the presence or the de-novo appearance or the volumetric increase in bronchiectasis is associated with Aspergillus colonization in COVID-19 patients [doi: 10.3390/diagnostics12071617]. Please cite this article and introduce these important aspects in the discussion section, describing if this imaging feature was observed also in the patient. However, as a recent study demonstrated, premortem studies underestimated the real burden of fungal and bacterial co-infections in COVID-19 patients compared to postmortem studies [doi: 10.3390/pathogens12070932].
We included the CT scans and both articles were cited and discussed
-Chapters 2 and 3 should be merged and “Case presentation” should be used as the title.
We merged these chapters together.
-Lines 140-141 seem to miss some references (possible suggestions include, doi: 10.3390/diagnostics12040846 and doi: 10.3390/diagnostics11112071).
Reference was added
-Line 165, as suggested above, explain if CAPA clinical diagnosis was excluded in your case and how it was performed.
-Please better explain why the Authors believe this is an interesting and unique case deserving of publication (for example, it should be mentioned in the conclusion that the persistence of SARS-CoV-2 in the tissues was demonstrated) and make it more engaging for the readers. Also, add a brief sentence explaining the importance of this report at the end of the introduction.
We rewrote the introduction and also the discussion.
Round 2
Reviewer 1 Report
None
Reviewer 3 Report
The Authors have addressed all raised points adequately